# Ln(III) Complexes Embedded in Biocompatible PLGA Nanoparticles as Potential Vis-to-NIR Optical Probes

**DOI:** 10.3390/molecules28052251

**Published:** 2023-02-28

**Authors:** Fabio Piccinelli, Silvia Mizzoni, Giorgia Zanella, Salvatore Calogero Gaglio, Massimiliano Perduca, Alessandro Romeo, Silvia Ruggieri, Chiara Nardon, Enrico Cavalli

**Affiliations:** 1Luminescent Materials Laboratory, DB, University of Verona, and INSTM, UdR Verona, Strada Le Grazie 15, 37134 Verona, Italy; 2Biocrystallography and Nanostructure Laboratory, Department of Biotechnology, University of Verona, Strada Le Grazie 15, 37134 Verona, Italy; 3Department of Computer Science, University of Verona, Strada Le Grazie 15, 37134 Verona, Italy; 4Department of Chemical Sciences, Life and Environmental Sustainability, University of Parma, Parco Area delle Scienze, 11/a, 43124 Parma, Italy

**Keywords:** ytterbium, neodimium, NIR luminescence, visible excitation, PLGA, nanoparticles, water dispersibility, cytotoxicity

## Abstract

In this contribution, we present the spectroscopic study of two NIR emitting hydrophobic heteroleptic (*R*,*R*)-Yb**L1**(tta) and (*R*,*R*)-Nd**L1**(tta) complexes (with tta = 2-thenoyltrifluoroacetonate and **L1** = N,N′-bis(2-(8-hydroxyquinolinate)methylidene)-1,2-(*R*,*R* or *S*,*S*)-cyclohexanediamine), both in methanol solution and embedded in water dispersible and biocompatible poly lactic-co-glycolic acid (PLGA) nanoparticles. Thanks to their absorption properties in a wide range of wavelengths extending from the UV up to the blue and green visible regions, the emission of these complexes can be effectively sensitized using visible radiation, which is much less harmful to tissues and skin than the UV one. The encapsulation of the two Ln(III)-based complexes in PLGA allows us to preserve their nature, making them stable in water and to test their cytotoxicity on two different cell lines, with the aim of using them in the future as potential bioimaging optical probes.

## 1. Introduction

Ln(III)-based complexes emitting in the NIR spectral region are considered potential candidates for applications in optical imaging and medical diagnostics [1,2,3]. The radiations belonging to the so-called “biological window” (650−1450 nm) are in fact scarcely absorbed and can deeply penetrate into cells and tissues with low scattering and reduced damage due to photobleaching [4]. Since the direct excitation of the Ln(III) ion gives rise to a weak luminescence emission, it is necessary to exploit the presence of at least an organic ligand, which can strongly absorb the light and transfer the excitation energy to the metal ion. Thanks to this ligand-to-metal energy transfer (LMET), also called *antenna* effect, the Ln(III) ion increases its emission intensity [5]. There is also another process which can increase the luminescence intensity of a Ln(III) ion, known as ligand-to-metal charge transfer (LMCT). This phenomenon is frequent in the case of Yb(III) [6,7], where the CT level can feed the ^2^F_5/2_ excited state of this ion.

As for the excitation wavelength, in the biological field, the use of UV light is strongly discouraged, thus the possibility of using *antennas* having suitable absorption properties in the visible regions or even in the NIR appears quite attractive. Another way to avoid the UV light is the recently developed technique of two-photon excitation (2PE), broadly employed in the luminescence of Ln(III)-based complexes [8,9,10]. This process involves the use of a 2P excitable *antenna*, which can absorb at the double of its maximum excitation wavelength, typically in the NIR spectral range.

Furthermore, a crucial aspect in biology is the water solubility and stability of compounds. When the species do not present these features in water, it is possible to overcome the problem by encapsulating them in a substance able to preserve the embedded compounds and at the same time to make them water dispersible and compatible. Among the most popular polymers, there is the poly lactic-co-glycolic acid (PLGA), known to have good biodegradability and biocompatibility properties and being Food and Drug Administration (FDA) and European Medicines Agency (EMA) approved for its administration in humans [11,12,13]. In previous works [14,15], after the encapsulation of the studied complexes, the resulting PLGA nanoparticles (PLGA NPs) showed the same luminescence efficiency evaluated for the naked species dissolved in organic solvents.

In this contribution, we present the spectroscopic characterization in solution of two heteroleptic complexes of Yb(III) and Nd(III), (Figure 1) bearing two chromophoric organic ligands (tta = 2-thenoyltrifluoroacetonate and **L1** = N,N′-bis(2-(8-hydroxyquinolinate)methylidene)-1,2-(*R*,*R* or *S*,*S*)-cyclohexanediamine), previously synthesized and characterized in the solid state [16]. Moreover, by encapsulating these species in PLGA nanoparticles, we made them water dispersible and compatible to evaluate the cytotoxicity for both, on two different human cancer cell lines.

The Yb(III)-based nano-systems proved to be stable enough at 37 °C and to have good biocompatibility, a feature required for future potential use as optical bioprobes.

## 2. Results

### 2.1. Spectroscopic Characterization of (R,R)-YbL1(tta) in Methanol Solution

The luminescence excitation spectrum of the Yb(III)-based complex in methanol solution (50 μM) with emission wavelength at 976 nm is reported in Figure 2A. The spectrum shows three intense features in the UV region (275, 310, and 350 nm) and a weaker and broad component extending into the visible region up to 550 nm, ascribable to transition pertaining to the organic ligand.

The NIR emission of the Yb(III) ion can then be efficiently excited in a wide wavelength range (250–550 nm), so upon excitation at three different wavelengths (340, 470, and 540 nm), relative to three different regions of the spectral range (UV, blue, and green region, respectively), the typical emission spectrum of a Yb(III) complex is observed (Figure 2B and Appendix A).

The spectrum involves a sharp zero-line peaking at 976 nm and the usual broad band, ranging from 900 to 1100 nm, related to the crystal field splitting of the ^2^F_7/2_ ground state of Yb(III).

### 2.2. Spectroscopic Characterization of (R,R)-NdL1(tta) in Methanol Solution

The Nd(III)-based complex in 50 μM methanol solution presents an excitation spectrum (emission wavelength 1058 nm) similar to the one of the Yb(III) counterpart (Figure 3A). In both cases, there is the same maximum close to 350 nm and a broad band in the visible region. This is obviously expected since the organic ligands are the same for the two complexes.

Likewise, the emission spectrum of the species was collected by exciting in both UV and visible regions, at 340 nm in the UV and at 470 nm in the blue (Figure 3B and Appendix A).

Upon excitation, the Nd(III) typical emission peaks from the ^4^F_3/2_ to the different ^4^I_J_ levels are observed (where J = 9/2, 11/2, and 13/2).

### 2.3. Preparation and Characterization of (R,R)-YbL1(tta) and (R,R)-NdL1(tta) Embedded in PLGA Nanoparticles

In order to make the two Ln(III)-based complexes dispersible and stable in water, we encapsulated them in PLGA nanoparticles, as previously reported in the literature, with a similar Yb(III)-based complex [15].

The encapsulation efficiency (EE) for these nanoparticles was found to be 44.1% ± 4.4 for the ones containing (*R*,*R*)-Yb**L1**(tta) and 63.1% ± 2.1 for the nanoparticles containing (*R*,*R*)-Nd**L1**(tta).

The size of each nano-formulation was estimated by comparing three different techniques: Dynamic Light Scattering (DLS), NanoTracking Analysis (NTA), and Atomic Force Microscopy (AFM) and results are summarized in Table 1

The DLS analysis of the samples gave size values of 138.8 ± 50.2 nm and 148.8 ± 40.9 nm, respectively, for the Yb and Nd complexes embedded in PLGA nanoparticles and both nano-formulations appeared monodisperse, as suggested by their Polydispersity Index (PDI) (0.070 ± 0.023 for PLGA Yb complex and 0.067 ± 0.025 for PLGA Nd complex). NTA size distribution of PLGA nanoparticles embedded with the Yb complex showed an average value of 129.9 ± 48.8 nm, where instead for the nanomaterial with Nd complex, it was of 100.5 ± 53.6. The two NTA values, considering the standard deviation, are in accordance with the DLS data. AFM analyses of the two nano-formulations yielded average diameters of 165.9 ± 38.9 nm and 179.1 ± 35.2 nm, respectively, for the Yb and Nd containing nano-formulations. The discrepancy between in solution data (DLS and NTA) and AFM data for PLGA nano-formulations can be ascribed to the formation of artifacts appearing during AFM measurements related to the deposition of the PLGA samples on the mica surface [17]. Figure 4 shows for both nano-formulations the presence of single well defined nanoparticles, homogeneous in shape and size, without highlighting macroscopical differences between the nanomaterials embedding the Yb(III) and Nd(III) complexes.

ζ-potential measurements of the two nano-formulations at physiological pH reveal a negatively charged surface [−17.8 ± 2.3 mV and −12.4 ± 3.1 mV for PLGA (Yb complex) and PLGA (Nd complex), respectively], thus improving the colloidal stability, as their aggregation is prevented due to electrostatic repulsion.

### 2.4. Spectroscopic Characterization of Water Dispersible NPs Containing (R,R)-YbL1(tta) and (R,R)-NdL1(tta)

The UV-visible spectra of the aqueous suspensions of Yb**L1**(tta)- and Nd**L1**(tta)-loaded PLGA NPs are reported in Figure 5A,B, respectively, together with those of the solutions of the complexes in methanol. The nature of the electronic transitions has been previously studied and discussed in detail [16].

This overlap, together with the one concerning the luminescence excitation spectra reported in Figure 6, suggests the presence of the intact species inside the polymer nanoparticles. In fact, even if not perfectly superimposable to the ones related to the naked species, both the absorption and excitation spectra show the same main peaks, albeit with some differences in intensity.

Furthermore, another proof of the preserved identity of both the Yb(III)- and Nd(III)-based complexes when encapsulated in the PLGA NPs, is given by the luminescence emission spectra (Figure 7), since they are fully consistent with the relative ones previously collected in methanol. Again, the emission spectra were collected by exciting in both UV and visible regions. More in detail, PLGA NPs containing the Yb(III) complex were excited at 340, 470, and 540 nm (Figure 7A and Appendix A), while the ones involving the Nd(III) counterpart were excited at 340 and 470 nm (Figure 7B and Appendix A).

### 2.5. Cytotoxicity Tests

The biological properties of the encapsulated complexes were preliminary evaluated against the two human cell lines A549 (non-small cell lung carcinoma) and HCT-116 (colon carcinoma). The screening involving the Nd derivative highlighted a not negligible cytotoxicity (EC_50_ < 10 µM). The water instability of its nano-formulation at 37 °C (Figure 8B) can account for the observed low cell viability.

Regarding the Yb(III)-based nanoparticles, both the enantiomers (*R*,*R* and *S,S*) were tested under the same experimental conditions. The similar but cationic chiral analogues containing a pyridine unit as a chromophore (Appendix A) [15] were likewise tested for comparison purposes. The obtained EC_50_ values are collected in Table 2.

Cell line being equal, the four Yb(III)-based complexes showed comparable cytotoxic properties, with higher EC_50_ values recorded on the A549 cell line.

Interestingly, both enantiomers showed similar cytotoxicity even though the (*R*,*R*) isomer seems to be slightly less toxic.

As for the charge of the embedded complexes, the class of compounds here reported is neutral, whereas the enantiomers recently published (named [Yb**L**(tta)_2_]CH_3_COO) [15] are cationic complexes with acetate as a counterion. The presence of the positive charge seems to affect the antiproliferative activity to a small extent.

Slightly smaller cytotoxicity was indeed recorded for the ionic complexes, likely due to a decreased capability to pass through the cell membrane.

## 3. Discussion

The possibility of exciting the Ln(III)-based complexes in a wavelengths range (285–550 nm) well extended into the visible region, making them of particular interest for applications in biology, as pointed out before. The broad band extended in the visible spectral range (up to 550–600 nm), is ascribed to the transition n → π* of the 8-hydroxyquinolinate ligand [18]. However, the excitation mechanisms are different for the two complexes: in the case of Nd**L1**(tta), the presence of several Nd(III) energy levels in the visible region is compatible with the occurring of a LMET process, whereas for the Yb(III) ion, the population of the ^2^F_5/2_ excited level presumably involves the formation of a LMCT state [16]. The difference in the sensitization mechanisms accounts for some differences (in shape and position) observed in the excitation spectra of the Nd(III)- and Yb(III)-based complexes.

The proof that both complexes preserve their composition when the powders of pure compounds are dissolved in solution or embedded in PLGA is given by spectroscopy. The presence of the excitation bands in the UV and visible regions calls for the involvement of both ligands (**L1** and tta) in the energy transfer process to the Ln(III) ions and consequently in the coordination to the metal centers. In fact, the luminescence of Yb(III) and Nd(III) can be sensitized upon excitation of **L1** (around 285 and 470 nm) or tta (340 nm) ligands. Following the Dexter’s energy transfer theory, the *antenna* effect works efficiently only when the donor–acceptor distance is short (where the donor is the ligand and the acceptor is the Ln(III) ion), in particular, when the exchange mechanism is involved [19]. This is the case of the herein analyzed complexes, where the triplet state of both ligands can efficiently feed the Ln(III) ions’ excited states.

An additional proof of the tta coordination in Yb**L1**(tta) is given by the comparison between the emission spectra of Yb**L1**(tta) and Yb**L1** (the latter complex lacks the tta ligand) (Appendix A). More in detail, upon excitation into the tta absorption band (at 340 nm), the Yb(III) emission intensity doubled in the Yb**L1**(tta) complex. Before evaluating their preliminary biological activity, however, it was necessary to test their stability not only at 20 °C, but also at 37 °C, the temperature employed for cytotoxicity tests, and in general for growing cells.

To do that, upon the assumption that the integrity of the luminescence properties of the complexes is closely related to the integrity of their structure, the emission area of the peaks of the two nano-formulations (at different temperatures) was checked every 24 h, up to 72 h (incubation time of cells after the addition of the nanoparticles embedding the complexes). From the graph in Figure 8A, it comes to light that PLGA nanoparticles trapping Yb**L1**(tta) are rather stable over time at both considered temperatures.

Unfortunately, the same trend has not been observed for NPs containing Nd**L1**(tta) which result to be not stable over time. In this case, there is a decrease in the intensity emission signal already after 24 h at 20 °C (Figure 8B). This drop is significant at 37 °C, where after the first 48 h, the typical emission peaks of Nd(III) are in practice not present. In order to provide a possible explanation about the different behavior of the two Ln(III)-based complexes, the spectroscopy of the species was also studied at 20 °C in three different environments (A-methanol; B-methanol/water = 2/1 and C-methanol/lactic acid). In both A and B cases, after 72 h, a decrease in the luminescence around 40% was detected regardless of the nature of the Ln(III) ion. In order to simulate the presence of the PLGA polymer around the metal complexes, we used a methanol solution containing an excess of lactic acid (one of the two monomers constituting the PLGA polymer). Interestingly, we noticed a major decrease in the luminescence intensity for Nd(III)-based complex after 30 min. The Nd(III) complex lost 55% of its original luminescence, whilst the Yb(III) one lost only 25% (Appendix A). Despite their similar stability in polar protic solvents (i.e., methanol and water), the interaction between the complexes and lactic acid seems to affect preferentially the integrity of the Nd(III) complex. In conclusion, this behavior could account for the different stability of the two metal complexes once embedded in the PLGA polymer.

The herein described complexes embedded in PLGA were tested against two human cancer cell lines over 72 h so to check the biological properties for a future potential use both ex vivo and in vivo, for instance as NIR-CPL (bio)-assays or trackers for CPL microscopy [15]. Both the chosen cell lines represent externally accessible neoplasms. The potential use as optical probes in non-small cell lung carcinoma (A549 cell line) and colon carcinoma (HCT-116) could occur via a switch on/off strategy. In fact, while most Schiff bases are stable under alkaline conditions, the imine bonds undergo hydrolysis (to yield the corresponding amine and aldehyde) in acid aqueous solutions. The tumor microenvironment is a complex milieu characterized by low nutrient levels, elevated interstitial fluid pressure, abnormal vascular network, and acidic pH with typical values ranging from 5 to 6.5 [20,21,22,23,24], thus leading to a possible bio-imaging selectivity. In light of the data collected in preliminary biological screening, Yb**L1**(tta) (as well as [Yb**L**(tta)_2_]CH_3_COO) embedded in PLGA NPs was poorly cytotoxic in both the investigated cell lines, in particular in the non-small cell lung carcinoma A549.

Anyway, it is important to highlight that both embedded Ln(III)-complexes are stable in the first hours, time required to perform bioimaging experiments, and so to be used as potential optical bioprobes, at least in cellulo experiments. The use of Nd(III) complexes as in vivo bioprobes is not recommended as they showed higher cytotoxicity.

## 4. Materials and Methods

Yb**L1**(tta) and Nd**L1**(tta) were synthesized and characterized as previously reported in the literature [16].

### 4.1. Preparation of YbL1(tta) and NdL1(tta) Loaded PLGA Nanoparticles

Polylactic-co-glycolic acid nanoparticles embedding Yb**L1**(tta) and Nd**L1**(tta) were prepared by the nanoprecipitation method, following the protocol described in Cavalli et al., 2022 [16]. First, 2 mg of PLGA (50:50) were resuspended in 1 mL of acetonitrile and 1 mg of Yb(III) or Nd(III) complex were added. Then, 50 µL of the previous suspension were added to 450 µL of 100 mM Glycine buffer pH 9, and later diluted to 3 mL with the same buffer. Samples were centrifuged for 40 min at 11,000 rpm and the nanoparticles were washed twice with a small amount of water, before resuspension in the final buffer; this colloidal solution was used for the spectroscopic measurements.

### 4.2. Encapsulation Efficiency Estimation

The entrapped amount of lanthanide complex was estimated by electronic absorption spectroscopy. Briefly, PLGA NPs were dissolved in acetonitrile, which induces a complete release of the intact metal complex in solution. The amount of the loaded complex into the NPs can be calculated from the calibration curve (Appendix A) by interpolation of the value of the absorbance at 340 nm of this solution.

Encapsulation efficiency (EE) was calculated applying the following Equation:$$\mathrm{EE}\;\left(\%\right)=\frac{Ln\;complex_{loaded}}{Ln\;complex_{fed}}\times 100$$
where *Ln complex_loaded_* is represented by the mass (mg) of Ln(III) complex encapsulated in PLGA and *Ln complex_fed_* is the mass (mg) of the starting complex.

### 4.3. Physicochemical Characterization of the Nano-Formulations

The nano-formulations obtained were characterized in terms of size (Dynamic Light Scattering–DLS, Nanoparticle Tracking Analysis–NTA, and Atomic Force Microscopy–AFM) and surface charge (ζ-potential).

Dynamic Light Scattering was performed using a Nano Zeta Sizer ZS, ZEN3600, version 7.10 (Malvern Instruments, Malvern, Worcestershire, UK) with all samples diluted 1 to 10 in Phosphate Buffer Saline (PBS) pH 7.4.

Nanoparticle Tracking Analysis (NTA) was performed using a Malvern NanoSight NS300 instrument (Worcestershire, UK) with PLGA nanoparticles embedding both Yb(III) or Nd(III) complex. Both samples were diluted 1:2 in MilliQ water and 1498 frames were recorded, divided in 3 runs of 50 s; for PLGA with the Yb complex, the camera level was set to 9, while for the nanoparticles embedding the Nd complex, the camera level was 12.

Samples for the AFM analysis were prepared loading 50 μL of each nanoparticle water suspension (diluted 1:1000 in MilliQ water) on a bracket covered by an inert mica surface. After 30 min of solvent evaporation, the analysis was performed using an NT-MDT Solver Pro atomic force microscope (Moscow, Russia) with NT-MDT NSG01 golden coated silicon tip in semi-contact mode, using different scanning frequencies (3–1 Hz) in order to produce optimized AFM images. The microscope was calibrated by a calibration grating (TGQ1 from NT-MDT) to reduce nonlinearity and hysteresis in the measurements. Finally, images were processed with the Scanning Probe Image Processor (SPIP™) program [25], and a statistical analysis considering 100 nanoparticles for each sample was performed to compare AFM results with DLS and Nanosight data.

Moreover, the colloidal stability of the samples was assessed measuring the ζ-potential of the nanoparticles resuspended in 10 mM NaClO_4_ pH 7.5, using a Nano Zeta Sizer ZS (ZEN3600, Malvern Instruments, Malvern, Worcestershire, UK); data were collected in triplicate and analyzed by ZetaSizer software.

### 4.4. Spectroscopic Characterization

The luminescence measurements were carried out by means of an Edinburgh FLS1000 spectrofluorometer equipped with both continuous and pulsed Xe lamp, a double excitation monochromator, a single emission monochromator, and a N_2_-cooled NIR extended photomultiplier for the detection of the emitted signal.

The preliminary spectroscopic measurements were carried out with a Fluorolog 3 (Horiba-Jobin Yvon) spectrofluorometer, equipped with a Xe lamp, a double excitation monochromator, a single emission monochromator (mod. HR320), and a photomultiplier in photon counting mode for the detection of the emitted signal.

All the spectra were measured at room temperature and corrected for the spectral responsiveness of the setup.

### 4.5. Cell Culture and Cell Viability Assay

Human colon carcinoma HCT-116 cells and non-small cell lung carcinoma A549 cells were obtained from American Type Culture Collection (Manassas, VA, USA) and grown in DMEM supplemented with 10% fetal bovine serum. Cells were grown in a humidified incubator with 5% CO_2_ at 37 °C. Cells were seeded in 96-well plates (volume = 100 μL; 5500 cells/well and 7000 cells/well for HCT-116 and A549, respectively) and grown to 70–75% confluence, followed by treatment with medium (control) or a chiral compound (previously encapsulated in PLGA and dissolved in the cell medium) at different concentrations in the micromolar domain under quadruplicate conditions. After a 72-h incubation at 37 °C, inhibition of cell proliferation was measured by the MTT assay, as previously described [26]. The cytotoxicity of the samples was quantified as the percentage of surviving cells compared to untreated cells. At least three MTT tests for each complex were carried out to evaluate the corresponding EC_50_ value, namely the concentration of the test complex inducing 50% reduction in cell number compared with control cultures (reported in Table 2). The polymer PLGA and the free metal chloride were tested as well (dissolved in cell culture medium), finding no cytotoxic effect under the same experimental conditions, except at concentrations > 100 µM for the latter.

## 5. Conclusions

In this paper, we presented two NIR emitting Ln(III)-based complexes (namely Yb**L1**(tta) and Nd**L1**(tta)), both excitable in a broad UV-visible spectral range. In particular, the likelihood of exciting in the visible region is strictly related to the nature of the chromophoric *antenna* where the conjugation of the hydroxiquinolinate ring is extended by the imine group. The Yb(III) luminescence is sensitized by means of a LMCT mechanism, whilst the Nd(III) one by the LMET process.

Moreover, we encapsulated the described Ln(III) complexes in the biodegradable and biocompatible PLGA NPs to make both the species dispersible and stable in water. It is worth highlighting that the imine pH-lability is paralleled with a good biological profile of the final compound in the case of Yb(III) ion. All the Yb(III)-based species tested showed less antiproliferative properties compared to, for instance, the clinically established drug cisplatin (EC_50_ ± SD = 3.6 ± 0.7 and 5.7 ± 0.2 µM against A549 and HCT-116 cell line, respectively) [27].

In conclusion, the chemical properties of the herein analyzed Ln(III) complexes, together with their biological features, allow them to be considered good candidates as optical probes in a convenient spectral range (vis-to-NIR). Furthermore, the embedded Yb(III)-based complexes can find possible applications also in in vivo experiments.

## Figures and Tables

**Figure 1 molecules-28-02251-f001:**
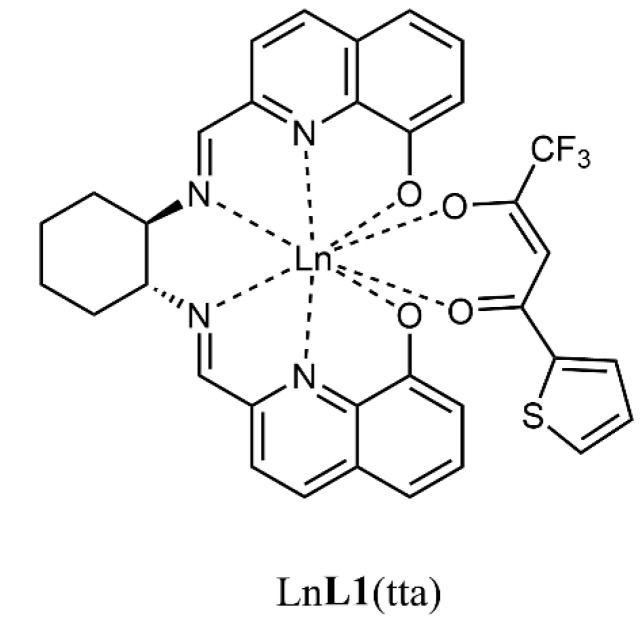
Molecular structure of the two complexes investigated in this contribution; Ln = Yb, Nd. The (*R*,*R*) enantiomer, chosen as representative, is here reported.

**Figure 2 molecules-28-02251-f002:**
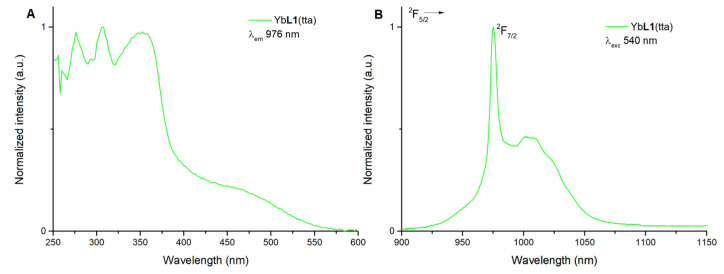
Luminescence excitation spectrum of (*R*,*R*)-Yb**L1**(tta) in methanol solution (λ_em_ = 976 nm) (**A**) and luminescence emission spectrum of (*R*,*R*)-Yb**L1**(tta) in methanol solution (λ_exc_ = 540 nm) (**B**). The spectra of (*S*,*S*) enantiomer (not reported) are superimposable.

**Figure 3 molecules-28-02251-f003:**
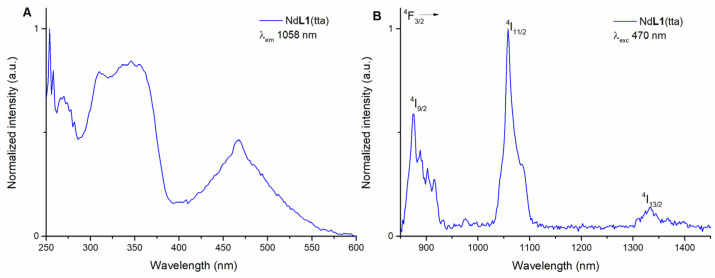
Luminescence excitation spectrum of (*R*,*R*)-Nd**L1**(tta) in methanol solution (λ_em_ = 1058 nm) (**A**) and luminescence emission spectrum of (*R*,*R*)-Nd**L1**(tta) in methanol solution (λ_exc_ = 470 nm) (**B**). The spectra of (*S*,*S*) enantiomer (not reported) are superimposable.

**Figure 4 molecules-28-02251-f004:**
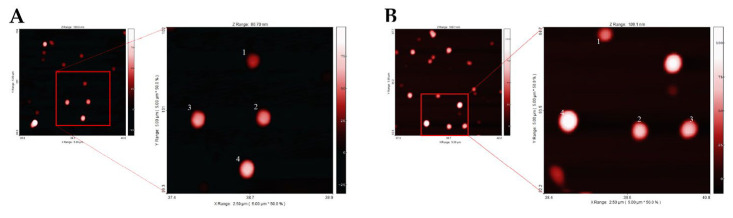
AFM images related to: PLGA(*R*,*R*)-Yb**L1**(tta): magnification of four nanoparticles with size close to AFM diameter mean (138 nm (1), 157 nm (2), 162 nm (3), 182 nm (4)) (**A**); PLGA(*R*,*R*)-Nd**L1**(tta): magnification of four nanoparticles with size close to AFM diameter mean (150 nm (1), 169 nm (2), 191 nm (3), 202 nm (4)) (**B**).

**Figure 5 molecules-28-02251-f005:**
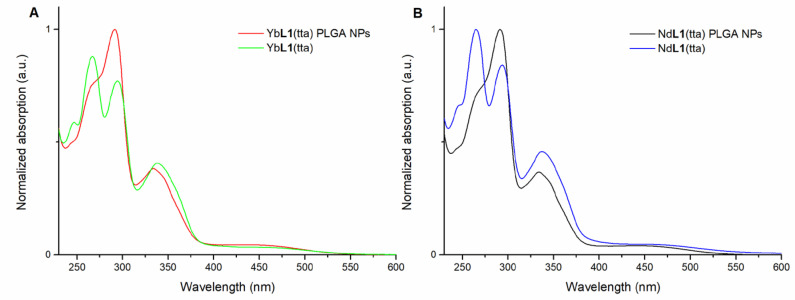
Overlap of electronic absorption spectra in the 230–600 nm range of: (*R*,*R*)-Yb**L1**(tta) in methanol and of PLGA NPs embedding (*R*,*R*)-Yb**L1**(tta) in water (**A**); (*R*,*R*)-Nd**L1**(tta) in methanol solution and of PLGA NPs embedding (*R*,*R*)-Nd**L1**(tta) in water (**B**). The spectra of (*S*,*S*) enantiomer (not reported) are superimposable.

**Figure 6 molecules-28-02251-f006:**
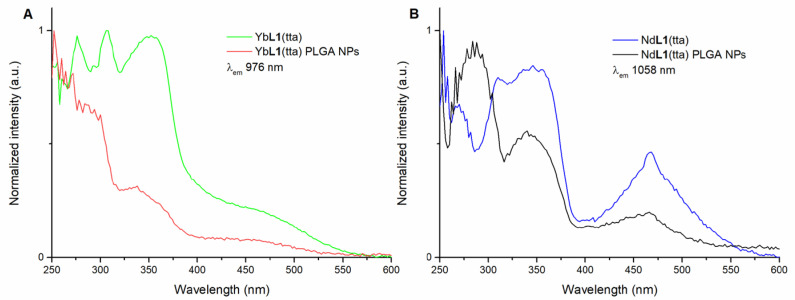
Overlap of luminescence excitation spectra of: (*R*,*R*)-Yb**L1**(tta) in methanol solution and (*R*,*R*)-Yb**L1**(tta) PLGA NPs in water (λ_em_ = 976 nm) (**A**); (*R*,*R*)-Nd**L1**(tta) in methanol solution and (*R*,*R*)-Nd**L1**(tta) PLGA NPs in water (λ_em_ = 1058 nm) (**B**). The spectra of (*S*,*S*) enantiomer (not reported) are superimposable.

**Figure 7 molecules-28-02251-f007:**
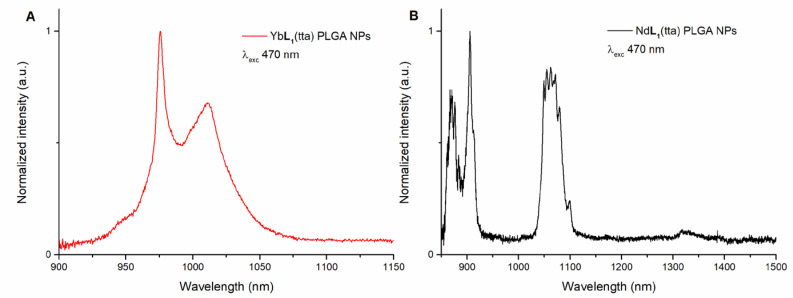
Luminescence emission spectra upon excitation at 470 nm of (*R*,*R*)-Yb**L1**(tta) (**A**) and (*R*,*R*)-Nd**L1**(tta) (**B**) both encapsulated in PLGA NPs. The spectra of (*S*,*S*) enantiomer (not reported) are superimposable.

**Figure 8 molecules-28-02251-f008:**
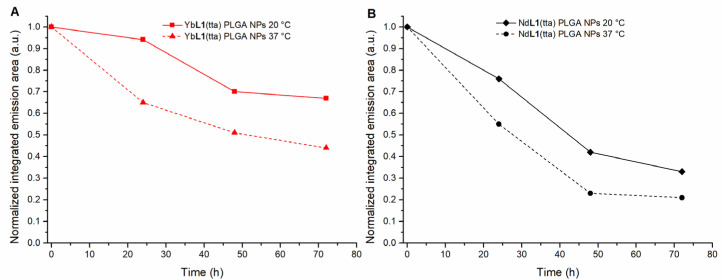
Evolution over the time (0 → 72 h) of the Ln(III) normalized integrated emission intensity for Yb**L1**(tta) (λ_exc_ = 470 nm, λ_em_ = 976 nm) (**A**) and for Nd**L1**(tta) (λ_exc_ = 470, λ_em_ = 1058 nm) (**B**), both embedded in PLGA, at two different temperatures (20 °C and 37 °C) in aqueous suspension.

**Table 1 molecules-28-02251-t001:** Comparison of size distribution by DLS data (Size and PDI), NTA, and AFM data of PLGA embedding Yb and Nd complexes. All measurements were performed in triplicate.

Sample	Size (nm)	PDI *	NTA Average Size (nm)	AFM Average Diameter (nm)
PLGA (*R*,*R*)-Yb**L1**(tta)	138.8 ± 50.2	0.070 ± 0.023	129.9 ± 48.8	165.9 ± 38.9
PLGA (*R*,*R*)-Nd**L1**(tta)	148.8 ± 40.9	0.067 ± 0.025	100.5 ± 53.6	179.1 ± 35.2

* Polydispersity Index.

**Table 2 molecules-28-02251-t002:** EC_50_ values (μM ± SD) of the investigated Yb(III)-based complexes against two human cancer cell lines.

Encapsulated Complex	A549 ^[a]^	HCT-116 ^[a]^
(*R*,*R*)-[Yb**L1**(tta)]	36 ± 1	15.2 ± 0.4
(*S*,*S*)-[Yb**L1**(tta)]	29.5 ± 0.8	12.4 ± 0.6
(*R*,*R*)-[Yb**L**(tta)_2_]CH_3_COO **^[b]^**	37.1 ± 0.5	18.4 ± 0.4
(*S*,*S*)- [Yb**L**(tta)_2_]CH_3_COO **^[b]^**	35 ± 1	16.5 ± 0.8

^**[a]**^ Each value represents the mean value of at least three-fold determinations after a 72-h treatment. ^**[b]**^ Yb(III) complexes previously discussed [15] similarly containing as chromophores two tta ligands and the **L** unit with a pyridine in place of the 8-hydroxyquinoline moiety (Appendix A).

## Data Availability

The data presented in this study are available in [*Chem. Eur. J.* **2022**, *28*, e202200574. https://doi.org/10.1002/chem.202200574 (accessed on 20 February 2023) and in *Results Chem.* **2022**, *4*, 100388. https://doi.org/10.1016/j.rechem.2022.100388 (accessed on 20 February 2023).

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
