# Peer review of "Ln(III) Complexes Embedded in Biocompatible PLGA Nanoparticles as Potential Vis-to-NIR Optical Probes"

_molecules, 2023, doi:10.3390/molecules28052251_

Round 1

Reviewer 1 Report

Manuscript by Silvia Ruggieri et al focuses on Nd(III) and Yb(III) complexes with NIR-luminescence, which have wide-ranging applications in the bioanalytical field. A number of papers of this group have been published in this field previously (Chem. Eur. J. 2022, 28, e202200574; Results in Chemistry 2022, 4, 100388). The investigated complexes are described by the authors earlier in Results in Chemistry 2022, 4, 100388. The only difference from the previous work is a study of the photophysical properties of the complexes embedded in PLGA nanoparticles. The nanoparticle study demonstrates the particle size. However, there is no information about the distribution of lanthanide complexes in the obtained nanoparticles. Also, the authors decided not to find an answer in this paper about the differences in the emission of nanoparticles with different lanthanide complexes. All this leaves the feeling that the authors in this article show only part of the research without taking it to a logical result. The study of photophysical properties and spectra largely repeat the data obtained earlier by the authors for the original complexes. On this basis, the novelty of this study lies only in the introduction of lanthanide complexes into the matrix of polymeric nanoparticles. The remarks to this work are the lack of structural characterization of the obtained complexes by XRD (which also applies to the previous works). This observation arises as there is no clear evidence of ligand coordination as indicated in the figure in the paper. Proof of the structure and composition of the complexes in solid state and in solution. Since the structural issue determines the specificity of the stability of the complexes, a comparison of the data by NMR is necessary. The second important observation (which can also be a consequence of the structure question) is the difference of the complexes when one lanthanide is replaced by another. Perhaps this is due to the different composition of the complexes or their stability in solution. As an assumption the diketonate anion could participate in the equilibrium in solution, which could also be established by NMR. Based on this, I recommend the authors to improve the paper and to include additional studies to establish the stability of the complexes in solution. I am not sure that such work can be done quickly, so I cannot recommend the article for publication in this version.

Reviewer 2 Report

Authors report the spectroscopic study on two NIR emitting metal complexes, both in methanol solution and in water dispersible PLGA nanoparticles. Manuscript is experimentally well performed and interesting for the potential use of Ln(III)-complexes as bioimaging optical probes.

I think the manuscript can be accepted for publication on Molecules but after the authors have clarified a very important aspect (major revision), concerning the stability of these metal complexes.

More in detail, considering the luminescence excitation spectra (paragraph 2.4), authors deduced that the intact species is present inside theNPs. Actually, Figures 5 and 6 show a different shape but, above all, the sentence is contrary to what the authors state later in the manuscript. Indeed, in the Discussion section, they affirm that the loss in luminescence is due to the breaking of the polymer assembly and the subsequent fast metal ion decomplexation and ligand hydrolysis.

Can authors clarify this aspect?

Reviewer 3 Report

Piccinelli et al. have described the spectroscopic studies of two lanthanide(III) complexes incorporated into biocompatible PLGA nanoparticles. Their results are new and interesting, and have been nicely and clearly described, especially from the applicative perspective of the water stable colloidal systems as bioimaging optical probes. That is why, I consider that this work deserves publication in Molecules, after the following minor point is considered by the authors: in Figure 1, it is enough to keep one single structure for both Ln(III) complexes, as they are identical, putting in the centre of the structure just Ln, where Ln = Yb or Nd.

Round 2

Reviewer 1 Report

I am satisfied with the changes made to the text of the manuscript and the authors' responses to the reviewer's comments. I am ready to reconsider my opinion and can recommend this work for publication.

Reviewer 2 Report

Authors replied satisfactorily to my doubts, clarifying all the aspects I paid attention to. I think the manuscript can be accepted for publication